# MicroRNA22-5p targets ten-eleven translocation and regulates estrogen receptor 2 expression in infertile women with minimal/mild endometriosis during implantation window

Li Xiao[1,2]◉, Tianjiao Pei[1,2]◉, Wei Huang[1,2]*, Min Zhou[1,2], Jing Fu[1,2], Jing Tan[1,2], Tingting Liu[1,2], Yong Song[1,2], Shiyuan Yang[1,2]

1 Department of Obstetrics and Gynecology, West China Second University Hospital of Sichuan University, Chengdu, Sichuan, People's Republic of China, 2 Key Laboratory of Birth Defects and Related Diseases of Women and Children (Sichuan University), Ministry of Education, Chengdu, Sichuan, People's Republic of China

◉ These authors contributed equally to this work.
* weihuang64@163.com

**Data Availability Statement:** All relevant data are within the manuscript and its Supporting Information files.

## Abstract

Based on microRNA (miR) microarray analysis, we previously found that miR22-5p expression is decreased in the mid-luteal endometrium of women with minimal/mild endometriosis. Bioinformatics analysis predicted that miR22-5p targets ten-eleven translocation (TET2) 3′-untranslated region. This study aimed to determine the regulation and roles of miR22-5p in the pathogenesis of minimal/mild endometriosis-associated infertility. MiR22-5p and TET2 expression in the mid-luteal endometrium from women with or without minimal/mild endometriosis was analyzed. After transfection with miR22-5p mimics or inhibitor, TET2 expression was analyzed by quantitative reverse transcription (RT-q) PCR, western blotting and immunohistochemistry. 5-Hydroxymethylcytosine was determined by immunofluorescence and dot blotting. Expression and promoter methylation of estrogen receptor 2 (ESR2) was measured by RT-qPCR and western blotting, and by bisulfite sequencing, respectively. We first established that miR22-5p expression decreased and TET2 expression increased in minimal/mild endometriosis during implantation window. TET2 was found to be a direct target of miR22-5p. MiR22-5p regulated the expression of ESR2, but did not directly affect methylation of its promoter region (−197/+359). Our results suggest that an imbalance in miR22-5p expression in the mid-luteal endometrium may be involved in minimal/mild endometriosis-associated infertility.

## Introduction

Endometriosis is characterized by the implantation and growth of endometrial tissue (glands and stroma) outside the uterine cavity. Women with endometriosis generally experience cyclical chronic pelvic pain, dyspareunia, and infertility, which significantly affect their quality of life. An estimated 30–50% of women with the disease are infertile, and 25–50% of infertile women are diagnosed as having endometriosis [1]. It has been suggested that endometriosis-associated infertility may be due to impaired pelvic anatomy, folliculogenesis, fertilization, and

**Funding:** This work was supported by a grant from the National Natural Science Foundation of China (No. 81370693). WH received the grant from the National Natural Science Foundation of China Grant number is 81370693 The full name of each funder: the National Natural Science Foundation of China URL of each funder website: http://www.nsfc.gov.cn/ The funders did not play any role in the study design, data collection and analysis, decision to publish, or preparation of the manuscript.

implantation [2–4]; however, the association between infertility and early-stage disease (minimal [stage I] and mild [stage II] endometriosis), in which no substantial pelvic anatomical changes are identified, remains controversial [4–6]. Endometrial receptivity is defined as a restricted period in the menstrual cycle during which the uterus is receptive to blastocyst attachment and implantation [7]. Innate pathology in the eutopic endometrium is suggested to contribute to implantation impairment in women with endometriosis with subfertility [8–10].

Progesterone resistance results in inadequate antagonism of estrogen action, increased inflammation, inadequate differentiation of the stroma, and remodeling of the endometrium, all of which can lead to a non-receptive endometrium for embryo implantation [11]. mRNA and protein levels of estrogen receptor 2 (ESR2) are significantly elevated in endometriotic cells as compared to normal stromal cells, whereas estrogen receptor 1 (ESR1), total progesterone receptor (PGR), and progesterone receptor B (PGR B) are repressed [12]. Increased ESR2 levels are directly related to *ESR2* promoter hypomethylation in endometriotic cells [12, 13]. ESR2 represses *ESR1* expression by directly binding to its promoter, which can result in progesterone resistance [12, 13]. However, the complete repertoire of ESR2 functions is believed to be more complicated.

MicroRNAs (miRNAs) are approximately 22-nucleotide non-coding RNAs that negatively regulate protein expression via translational inhibition or mRNA degradation. Emerging data suggest that dysregulation of miRNA expression may be implicated in the development and pathogenesis of endometriosis [14]. Using microarray-based miRNA profiling, we previously found that 66 mature miRNAs were differentially expressed (54 up- and 12 downregulated) in mid-luteal eutopic endometrium from women with minimal/mild endometriosis as compared with normal endometrial tissues [15]. Specifically, miR-196a upregulated MEK/ERK signaling and mediated repressed PGR expression and decidualization of endometrial stromal cells (ESCs) from eutopic endometrium with endometriosis [15]. Likewise, upregulation of miR-194-3p in eutopic endometrium inhibited PGR expression and ESC decidualization in endometriosis, which hinders fertility by repressing the levels of PGR and decidualization in the eutopic endometrium [16]. We have also shown that miR23a and miRNA23b are downregulated in endometriosis and they upregulate several unidentified genes required for Steroidogenic factor 1 (SF-1) expression in ESCs [17].

Bioinformatics analysis of a downregulated miRNA, miR22-5p, revealed a binding site in the 3′-untranslated region (3′-UTR) of ten-eleven translocation 2 (TET2) [18]. The TET family encodes enzymes responsible for the oxidation of 5-methylcytosine (5mC) to 5-hmC in DNA demethylation. Genome-wide analysis of endometriomas revealed significantly low *TET* gene expression associated with high 5-hmC levels upon *in-vitro* decidualization, suggesting a unique epigenetic regulation in these ectopic tissues [19]. However, how TET2 affects minimal/mild endometriosis-related infertility and the specific mechanism are unknown.

The aim of the present study was to evaluate miR22-5p expression and the relationship between miR22-5p and TET2 expression in mid-luteal eutopic endometrium of infertile women with and without minimal/mild endometriosis using tissues and primary ESCs.

## Material and methods

### Study population

This study was approved by the Medical Research Review Board of West China Second University Hospital of Sichuan University, and written informed consent for participation was obtained from all participants. In total, 50 infertile women aged 22–34 years old with regular menstrual cycles were enrolled in this study between January 2015 and May 2018. Normal endometrium was obtained from 24 infertile women without endometrial pathology, and eutopic endometrium was obtained from 26 infertile women with a laparoscopic and histological diagnosis of stage I–II

endometriosis according to the revised American Fertility Society classification system (Table 1). Participants with adenomyosis, leiomyomas, endometrial hyperplasia, genital tumors, acute pelvic inflammatory disease, or receiving hormonal treatment within the previous three months were excluded. Secretory-phase endometrial tissues, assessed based on the timing of the last menstrual period and histological analysis, were used in the study.

## Primary cell culture and transfection

ESCs were isolated from eutopic endometrium (n = 3) and normal endometrium (n = 3) as described previously. Isolated ESCs were cultured in DMEM/F12 (1:1) supplemented with 10% fetal bovine serum (FBS) at 37°C. 293T human embryonic kidney cells were obtained from Sichuan University, Chinese University of Hong Kong Joint Laboratory for Reproductive Medicine, and were maintained in DMEM supplemented with 10% FBS. When cells reached 80% confluency, they were trypsinized and seeded into 6-well plates at $1.0 \times 10^5$ cells/mL. When the cells reached 30–40% confluency, they were transfected with Hsa-miR22-5p mimics or inhibitor (100 nM, Guangzhou RiboBio, Guangzhou, China), using Lipofectamine 3000 transfection reagent (Invitrogen, Carlsbad, CA), according to the manufacturer's protocol. After 48–72 h of culture, the cells were harvested and collected for mRNA isolation or protein extraction. All experiments were performed in triplicate.

## RNA extraction and quantitative reverse-transcription (RT-q)PCR

Total RNA was extracted from endometrial tissues and primary cultured ESCs using TRIzol reagent (Life Technologies, Carlsbad, CA), according to the manufacturer's protocol. RNA quality and purification were analyzed using a NanoVue Plus spectrophotometer (Healthcare Bio-Sciences AB, Uppsala, Sweden). cDNA was synthesized from purified total RNA using a PrimeScript RT reagent kit (TaKaRa Biotechnology, Dalian, China). Primer sequences for *TET2*, *ESR1*, *ESR2*, and *GAPDH* (Sango Biotech, Shanghai, China) are listed in Table 2. qPCRs were run using SYBR Green real-time PCR Master Mix (Toyobo, Osaka, Japan) on an Applied Biosystems 7900 Real-time PCR Detection System (ABI, Foster City, CA). The thermal cycles were 95°C for 20 s followed by 40 cycles of 95°C for 10 s and 60°C for 20 s. The specificity of PCR products was confirmed by dissociation curve analysis. *GAPDH* was used as an endogenous control to normalize target gene expression, and relative expression was calculated using the $2^{-\Delta\Delta Ct}$ method. For the quantitation of mature miRNAs, miRNA RT-qPCR was conducted using specific primers for miR22-5p and U6 small nuclear RNA (as an internal control) from the Bulge-Loop qRT-PCR Primer Set (Guangzhou RiboBio, Guangzhou, China), according to the manufacturers' protocol. All experiments were repeated three times.

## Western blot analysis

Total protein was extracted using radio immunoprecipitation lysis buffer (P0013B, Beyotime Biotechnology, Shanghai, China) according to the manufacturer's instructions. Protein concentrations were determined using a bicinchoninic acid assay kit (Beyotime Biotechnology). Proteins (30 μg) from each sample were separated by 10% sodium dodecyl sulfate–polyacrylamide gel electrophoresis and were transferred to polyvinylidene fluoride membranes (Millipore, Billerica, MA). The membranes were blocked in 5% defatted milk at room temperature for 1 h. Then, the membranes were incubated with mouse anti-human TET2 (1:400; ab94580, Abcam, Cambridge, UK), polyclonal rabbit anti-human ESR2 (1:400; ab3577, Abcam), polyclonal mouse anti-β-actin (1:30000; bs-2188R, Bioss, Beijing, China) antibodies at 4°C overnight and then with horseradish peroxidase–conjugated secondary anti-mouse/rabbit antibody at room temperature for 1 h. Proteins bands were visualized using an enhanced

**Table 1. Details of patient samples used in this study.**

| Code | Age | Menstrual stage | Indication/diagnosis | Assay |
|------|-----|-----------------|----------------------|-------|
| 1 | 25 | Secretory | Minimal/mild endometriosis | IHC |
| 2 | 25 | Secretory | Minimal/mild endometriosis | IHC, Q |
| 3 | 28 | Secretory | Minimal/mild endometriosis | IHC, Q |
| 4 | 31 | Secretory | Minimal/mild endometriosis | Q |
| 5 | 34 | Secretory | Minimal/mild endometriosis | Q |
| 6 | 32 | Secretory | Minimal/mild endometriosis | Q |
| 7 | 26 | Secretory | Minimal/mild endometriosis | Q |
| 8 | 25 | Secretory | Minimal/mild endometriosis | Q |
| 9 | 23 | Secretory | Minimal/mild endometriosis | Q |
| 10 | 31 | Secretory | Minimal/mild endometriosis | Q |
| 11 | 33 | Secretory | Minimal/mild endometriosis | Q |
| 12 | 28 | Secretory | Minimal/mild endometriosis | Q |
| 13 | 23 | Secretory | Minimal/mild endometriosis | Q |
| 14 | 34 | Secretory | Minimal/mild endometriosis | Q |
| 15 | 27 | Secretory | Minimal/mild endometriosis | WB |
| 16 | 27 | Secretory | Minimal/mild endometriosis | WB |
| 17 | 26 | Secretory | Minimal/mild endometriosis | WB |
| 18 | 28 | Secretory | Minimal/mild endometriosis | WB |
| 19 | 31 | Secretory | Minimal/mild endometriosis | WB |
| 20 | 32 | Secretory | Minimal/mild endometriosis | WB |
| 21 | 34 | Secretory | Minimal/mild endometriosis | PCC |
| 22 | 26 | Secretory | Minimal/mild endometriosis | PCC |
| 23 | 26 | Secretory | Minimal/mild endometriosis | PCC |
| 24 | 29 | Secretory | Minimal/mild endometriosis | PCC |
| 25 | 24 | Secretory | Minimal/mild endometriosis | PCC |
| 26 | 26 | Secretory | Minimal/mild endometriosis | PCC |
| 27 | 26 | Secretory | Peritubal adhesion | Q |
| 28 | 28 | Secretory | Peritubal adhesion | Q |
| 29 | 25 | Secretory | Pelvic adhesion | Q |
| 30 | 30 | Secretory | Peritubal adhesion | Q |
| 31 | 31 | Secretory | Peritubal adhesion, mesosalpinx cyst | Q |
| 32 | 25 | Secretory | Peritubal adhesion, mesosalpinx cyst | Q |
| 33 | 28 | Secretory | Peritubal adhesion, mesosalpinx cyst | Q |
| 34 | 28 | Secretory | Peritubal adhesion, mesosalpinx cyst | Q |
| 35 | 24 | Secretory | Peritubal adhesion | Q |
| 36 | 29 | Secretory | Peritubal adhesion | Q |
| 37 | 24 | Secretory | Peritubal adhesion | IHC, Q |
| 38 | 31 | Secretory | Peritubal adhesion | IHC |
| 39 | 33 | Secretory | Peritubal adhesion | IHC |
| 40 | 32 | Secretory | Peritubal adhesion | WB |
| 41 | 25 | Secretory | Peritubal adhesion | WB |
| 42 | 26 | Secretory | Peritubal adhesion | WB |
| 43 | 28 | Secretory | Peritubal adhesion | WB |
| 44 | 24 | Secretory | Pelvic adhesion | WB |
| 45 | 26 | Secretory | Pelvic adhesion | PCC |
| 46 | 24 | Secretory | Pelvic adhesion | PCC |
| 47 | 22 | Secretory | Peritubal adhesion | PCC |

(*Continued*)

**Table 1.** (Continued)

| Code | Age | Menstrual stage | Indication/diagnosis | Assay |
|------|-----|-----------------|----------------------|-------|
| 48 | 28 | Secretory | Peritubal adhesion | PCC |
| 49 | 31 | Secretory | Peritubal adhesion | PCC |
| 50 | 32 | Secretory | Peritubal adhesion | PCC |

chemiluminescence system (Millipore) and were analyzed with ImageJ 2X (National Institutes of Health, Bethesda, MD). Protein levels were normalized to that of β-actin.

## Luciferase reporter assay

The base sequences of *TET2* 3′-UTR fragments were identical to the sequences in the NCBI public bioinformation resource (www.ncbi.nlm.nih.gov/gene/54790). MiR22-5p targets were predicted using TargetScan (http://www.targetscan.org) and microRNA.org (http://microrna.org). Three potential miR22-5p target sites were identified in the 3′-UTR of the *TET2* mRNA sequence. Reporter genes were constructed by PCR amplification, gel purification, and restriction digest of the *TET2* 3′-UTR. Three wild-type (WT) *TET2* 3′-UTR fragments each containing one of the predicted miR22-5p-binding sites were cloned into the pmiR-RB-REPORT vector (Promega, Fitchburg, WI). The constructed reporter plasmids were designated TET2 3′-UTR WT1, TET2 3′-UTR WT2, and TET2 3′-UTR WT3. The differential expressed TET2 3'-UTR WT1, mutation of TET2 3'-UTR WT1 (TET2 3'-UTR M1) was also designated. 293T cells were transfected with the reporter plasmids and 50 nM of miR22-5p mimic or miRNA negative control (Guangzhou RiboBio, Guangzhou, China). After 48 h, the cells were harvested and luciferase activity was measured using a dual-luciferase reporter assay system (Promega) according to the manufacturer's protocol. All transfections and assays were performed three times, with six technical replicates.

## Immunofluorescence

Endometrial cells ($25 \times 10^3$) were seeded on glass coverslips in 24-well plates. After treatment, the cells were fixed in 4% paraformaldehyde in 1× PBS for 15 min, washed in PBS, and treated with 0.2% Triton X-100 in PBS for 15 min. Permeabilized cells were denatured with 2 N HCl for 15 min and neutralized with 100 mM Tris-HCl (pH 8.5) for 10 min. Proteins were blocked in 1% BSA in PBS for 30 min and then, the cells were incubated with rabbit anti-human 5-hmc antibody (1:100) and mouse anti-human TET2 antibody (1:200) at room temperature for 2 h, followed by Alexa Fluor 488-labeled anti-mouse antibody (Life Technologies). After washing, the cells were counterstained with 4′,6-diamidino-2-phenylindole.

## Dot blot assay

Genomic DNA was extracted from primary cells using a DNeasy Blood & Tissue Kit (Qiagen) according to the manufacturer's protocol. DNA samples were diluted with 2 N NaOH and 10

**Table 2. Primers used for RT-qPCR.**

| Gene | Forward primer (5′-3′) | Reverse primer (5′-3′) |
|------|------------------------|------------------------|
| TET2 | ATACCCTGTATGAAGGGAAGCC | CTTACCCCGAAGTTACGTCTTTC |
| ESR1 | GAAAGGTGGGATACGAAAAGACC | GCTGTTCTTCTTAGAGCGTTTGA |
| ESR2 | AGCACGGCTCCATATACATACC | TGGACCACTAAAGGAGAAAGGT |
| GAPDH | TGCACCACCAACTGCTTAGC | GGCATGGACTGTGGTCATGAG |

mM Tris Cl (pH 8.5) and blotted onto a nitrocellulose membrane. After baking at 80°C for 30 min and blocking in 5% nonfat milk at room temperature for 1 h, the membrane was incubated with a polyclonal rabbit anti-human 5-hmC antibody (Active Motif 39769, 1:10,000) at 4°C overnight. 5-hmC was visualized using chemiluminescence. The membranes were stained with methylene blue to assess equal DNA loading.

## Immunohistochemistry

Tissue was embedded in paraffin, cut into 5-μm sections, and mounted onto gelatin-coated slides. Sections were dried at 37°C overnight, deparaffinized in xylene, and rehydrated through a graded ethanol series. The slides were immersed in citrate antigen retrieval buffer (pH 6) at 120°C for 10 min to retrieve the epitopes, incubated with 3% $H_2O_2$ for 10 min to block endogenous peroxidase activity after cooling, blocked with 10% normal goat serum for 30 min, and incubated with the primary antibody (TET2 1:200, 5-hmc 1:200) at 4°C overnight. Biotinylated secondary antibody and streptavidin-peroxidase conjugate were applied according to the manufacturer's instructions (Beijing Zhongshan Biotech, Beijing, China). Immunoreactivity was visualized with diaminobenzidine, and the sections were counterstained with hematoxylin and mounted. Isotype controls were performed with matched concentrations of mouse IgG for TET2 and rabbit IgG for 5-hmC.

## Bisulfite modification and sequencing

Genomic DNA was extracted from miR22-5p inhibitor- and control-transfected primary ESCs (n = 3) from infertile women without endometriosis using the DNeasy Tissue Kit (Qiagen) and was used for bisulfite modification and sequencing analysis (Sangon Biotech, Shanghai, China). Three microliters of bisulfite-modified DNA was PCR-amplified a reaction volume of 50 μl, using the following primers for *ESR2*: forward: 5′–ATTATTTTTGTGGGTGGATTAG GAG–3′, and reverse: 5′–AACCCCTTCTTCCTTTTAAAAACC–3′. Thermal cycles were as follows: 98°C for 4 min, 20 cycles of denaturation at 94°C for 45 s, annealing at 66°C for 45 s, and elongation at 72°C for 1 min, and 20 cycles of denaturation at 94°C for 45 s, annealing at 56°C for 45 s, and elongation at 72°C for 1 min, and finally, 72°C for 8 min. PCR products (166 bp) were gel-purified and cloned into the pUC18-T vector (Sangon Biotech). Following transformation, ten clones with the correct insert were randomly picked for each PCR product and were sequenced using an Applied Biosystems 3730XL instrument.

## Statistical analysis

Statistical analysis was performed using SPSS version 18.0 (IBM Corp., USA). All data were expressed as the mean ± SD. Means of two groups were compared using Student's *t*-test. $P < 0.05$ was considered statistically significant (two-tailed).

# Results

## MiR22-5p and TET2 expression signature discriminates eutopic endometrium of mild/minimal endometriosis from normal endometrium

In our previous microarray-based miRNA profiling analysis of mid-luteal endometrium from women with minimal and mild endometriosis, we observed a marked downregulation of miR22-5p [15]. To confirm these findings in this study, we subjected 13 endometrial tissues of mild/minimal endometriosis and 11 normal control tissues to RT-qPCR. MiR22-5p expression was significantly lower in eutopic endometrium of minimal/mild endometriosis than in normal tissues (Fig 1A). Next, we investigated the biological significance of miR22-5p

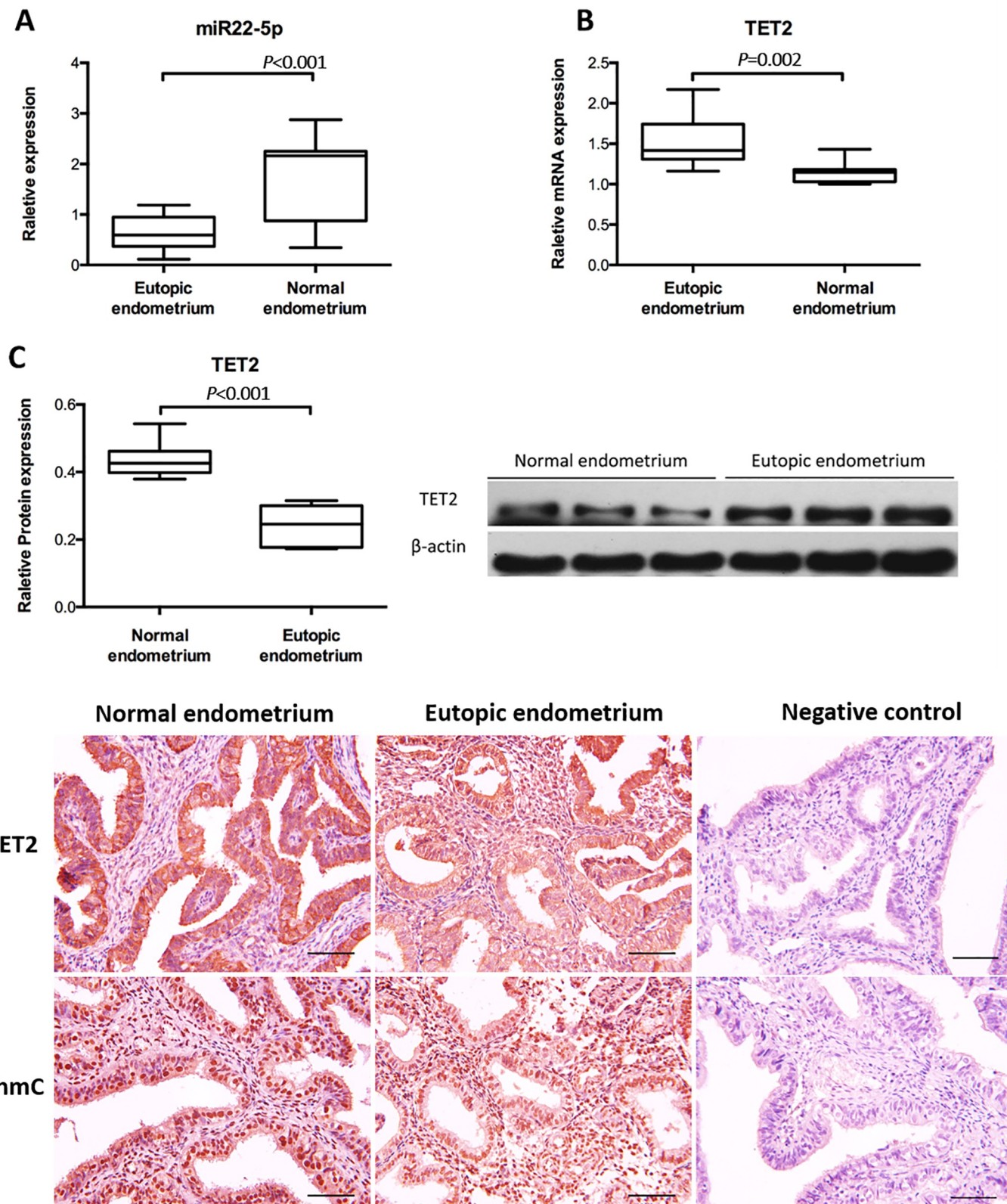

**Fig 1. MiR22-5p, TET2, and 5-hmC expression in endometrium of minimal/mild endometriosis and normal endometrium.** (A) RT-qPCR analysis of miR22-5p and (B) *TET2* expression in endometrium of minimal/mild endometriosis (n = 13) and normal endometrium (n = 11). miRNA expression was normalized to U6 snRNA expression, and mRNA expression was normalized to *GAPDH* expression. (C) Western blot analysis of TET2 expression in mild endometriosis (n = 6) and in normal endometrium (n = 5), normalized to GAPDH expression. (D) Representative Immunohistochemistry images showing TET2 and 5-hmC in the cytoplasmic and nuclear regions of stromal and epithelial cells in endometrium of mild endometriosis and normal endometrium (n = 3). Black arrows indicate positive staining. Scale bars, 200 μm. Data represent the mean ± SD and were analyzed by Student's *t*-test.

downregulation in the regulation of target *TET2* mRNAs. Compared to endometrium from women without endometriosis, *TET2* expression was significantly higher in endometrium from women with minimal/mild endometriosis during the secretory phase (Fig 1B). TET2 protein levels were significantly higher in endometrium from women with minimal/mild endometriosis than in normal endometrium as indicated by western blot results (Fig 1C). Immunohistochemistry revealed that TET2 was strongly expressed in both the nucleus and the cytoplasm of epithelial and stromal cells of eutopic endometrium of endometriosis, whereas normal endometrium exhibited weak to moderate expression (Fig 1D). The above results suggested a correlation between miR22-5p and TET2 in minimal/mild endometriosis. The DNA pyrimidine nitrogen base 5-hmC was strongly expressed in both the nucleus and the cytoplasm of epithelial and stromal cells of eutopic and normal endometrium (Fig 1D).

## MiR22-5p regulates TET2 expression in primary ESCs

We examined the impact of miR22-5p on TET2 expression by transfecting primary ESCs with miR22-5p mimics and inhibitor. Successful miR22-5p mimic and inhibitor transfection was validated by a significant increase (0.85 ± 0.15 vs. 5189.65 ± 3062.38; $P < 0.001$; Fig 2A) and decrease (0.98 ± 0.22 vs. 0.17 ± 0.09; $P < 0.001$; Fig 2A), respectively, in miR22-5p in ESCs. Upon miR22-5p mimic transfection, we observed significant decreases in the *TET2* mRNA (1.03 ± 0.12 vs. 0.51 ± 0.24, $P < 0.001$, Fig 2B) and protein (1.05 ± 0.20 vs. 0.12 ± 0.04, $P = 0.001$, Fig 2C and 2E) levels. Accordingly, following transfection with miR22-5p inhibitor, we observed significant increases in the *TET2* mRNA (1.07 ± 0.17 vs. 1.52 ± 0.28, $P < 0.001$, Fig 2B) and protein (1.53 ± 0.14. vs. 1.02 ± 0.19, $P = 0.02$, Fig 2D and 2E) levels.

Next, we investigated 5-hmC expression in genomic DNA of ESCs following transfection with miR22-5p mimics and inhibitor by dot blot assays (Fig 2F). Treatment with the miR22-5p mimics and inhibitor decreased and increased global 5-hmC, respectively. Consistent with the western blotting and DNA methylation dot blot assay results, immunofluorescence revealed that the expression of TET2 and 5-hmC was decreased in miR22-5p mimics-treated cells, whereas they were overexpressed in miR22-5p inhibitor-treated cells (Fig 3).

## MiR22-5p directly targets the *TET2* 3′-UTR

We used a luciferase reporter assay to determine whether TET2 regulation was mediated by direct binding of miR22-5p to its 3′-UTR (Fig 4A). 293T cells cotransfected with a reporter plasmid and miR22-5p mimics showed significantly decreased luciferase activity as compared to cells transfected with the reporter plasmid alone only for the TET2 3′-UTR WT1 construct, among the three wild-type constructs evaluated (Fig 4B). Direct interaction of miR22-5p with TET2 was confirmed by using a luciferase reporter harboring a point mutation (Fig 4C).

## MiR22-5p affects ESR2 mRNA and protein expression

To investigate the impact of miR22-5p on ESR2 expression, we assessed primary ESCs following transfection with miR22-5p mimics or its inhibitor. *ESR2* mRNA expression was significantly downregulated after transfection with miR22-5p mimics (1.11 ± 0.30 vs. 0.42 ± 0.35,

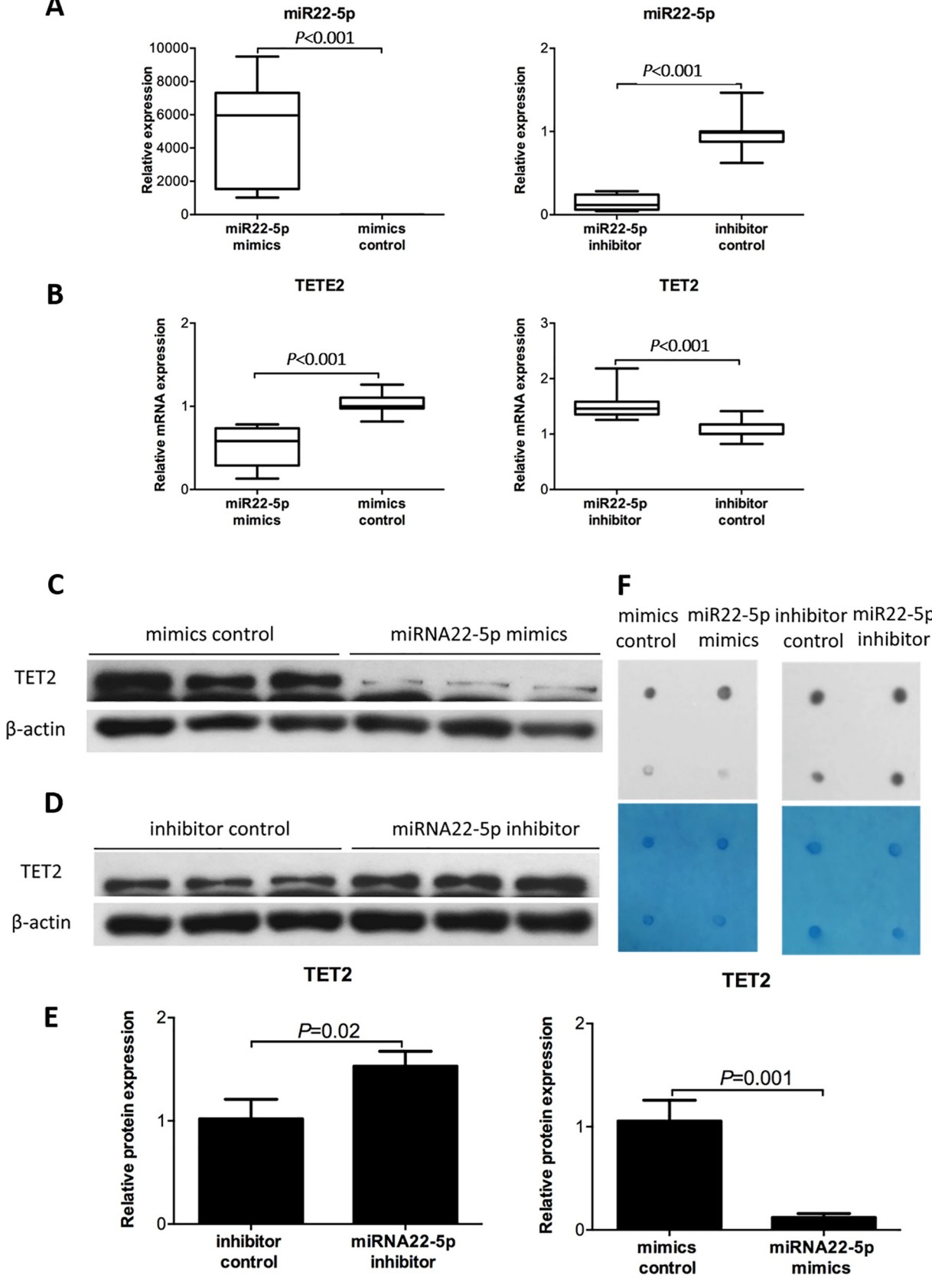

**Fig 2. MiR22-5p regulates TET2.** (A) Expression of miR22-5p in ESCs treated with miR22-5p mimics or inhibitor and their negative control. (B–E) mRNA and protein expression of TET2 in ESCs following treatment with miR22-5p mimics and inhibitor as measured by RT-qPCR and western blotting. (F) Dot blot analysis of TET2 expression in ESCs following treatment with miR22-5p mimics and inhibitor.

$P < 0.001$, Fig 5A), and significantly upregulated after treatment with the inhibitor ($0.84 \pm 0.20$ vs. $1.83 \pm 0.41$, $P < 0.001$, Fig 5A). There was no difference in *ESR1* expression following either treatment. Accordingly, the *ESR1/ESR2* mRNA ratio was significantly increased after transfection of miRNA22-5p mimics ($1.07 \pm 0.33$ vs. $6.18 \pm 0.33$, $P = 0.019$, Fig 5A) or inhibitor ($1.142 \pm 0.48$ vs. $0.68 \pm 0.29$, $P < 0.001$, Fig 5A). ESR2 protein expression was significantly decreased in cells treated with miRNA22-5p mimics ($0.87 \pm 0.11$ vs. $0.43 \pm 0.15$, $P < 0.001$, Fig 5B) and increased in cells treated with inhibitor ($0.82 \pm 0.31$ vs. $1.47 \pm 0.21$, $P < 0.001$, Fig 5B), as demonstrated by western blot analysis.

### MiR22-5p affects the DNA methylation status of the *ESR2* promoter region expression in primary endometrial cells

To investigate the role of miR22-5p in DNA methylation further, we assessed the transfected ESCs by bisulfite sequencing. We identified and approximately 550-bp classic CpG island (–197/+359) within the *ESR2* promoter and the downstream untranslated exon 0N region. The CpG methylation status after treatments is shown in Fig 6A. The methylation status within this region after transfected with miRNA 22-5p inhibitor or controls was very low and it was not statistically significant (Fig 6B).

### Discussion

Endometriosis is an estrogen-dependent chronic inflammatory disease that contributes to cyclical chronic pain and infertility in reproductive women. Recent research has focused on microRNAs because of their roles in regulating epigenetic changes involved in the pathophysiology of infertility due to endometriosis. Previously, utilizing microRNA array analysis, we identified miR22-5p as one of 12 downregulated miRNAs in eutopic mid-luteal endometrium of minimal/mild endometriosis [15]. In the present study, we validated that miR22-5p expression was decreased in eutopic endometrium of minimal/mild endometrium during the mid-luteal phase. Utilizing a primary ESC model, we found that miR22-5p expression was inversely related to TET2 and 5-hmC expression. MiR22-5p regulated ESR2 expression, but did not directly affect ESR2 promoter methylation. These results support the hypothesis that miRNAs, and *in casu*, miR22-5p, contributes to the pathophysiology of endometriosis and may be a useful therapeutic target.

Women with minimal/mild endometriosis are significantly less likely to achieve pregnancy than those with tubal factor infertility [20]. Endometriosis likely is the most common cause of endometrial receptivity defects, especially in cases of minimal/mild endometriosis, in which the loss of fertility cannot be explained by mechanical reasons [21]. How the endometrium in women with minimal/mild endometriosis is resistant to embryo implantation remains unknown. Maciejak et al. reported upregulation of miR22-5p in the plasma and serum following acute myocardial infarction as a novel diagnostic biomarker [22]. A direct relationship between miR22-5p and TET2 expression has been demonstrated in K562 cells, and in AML cells, low TET2 expression is related to proliferation [23]. However, it has been widely reported that miR-22 negatively regulates TET2 expression and that its overexpression closely phenocopies many of the characteristics observed upon TET2 inactivation both *in vitro* and *in vivo* [24]. MiR-22 contributes to the inactivation of TET2 and other TET family members in

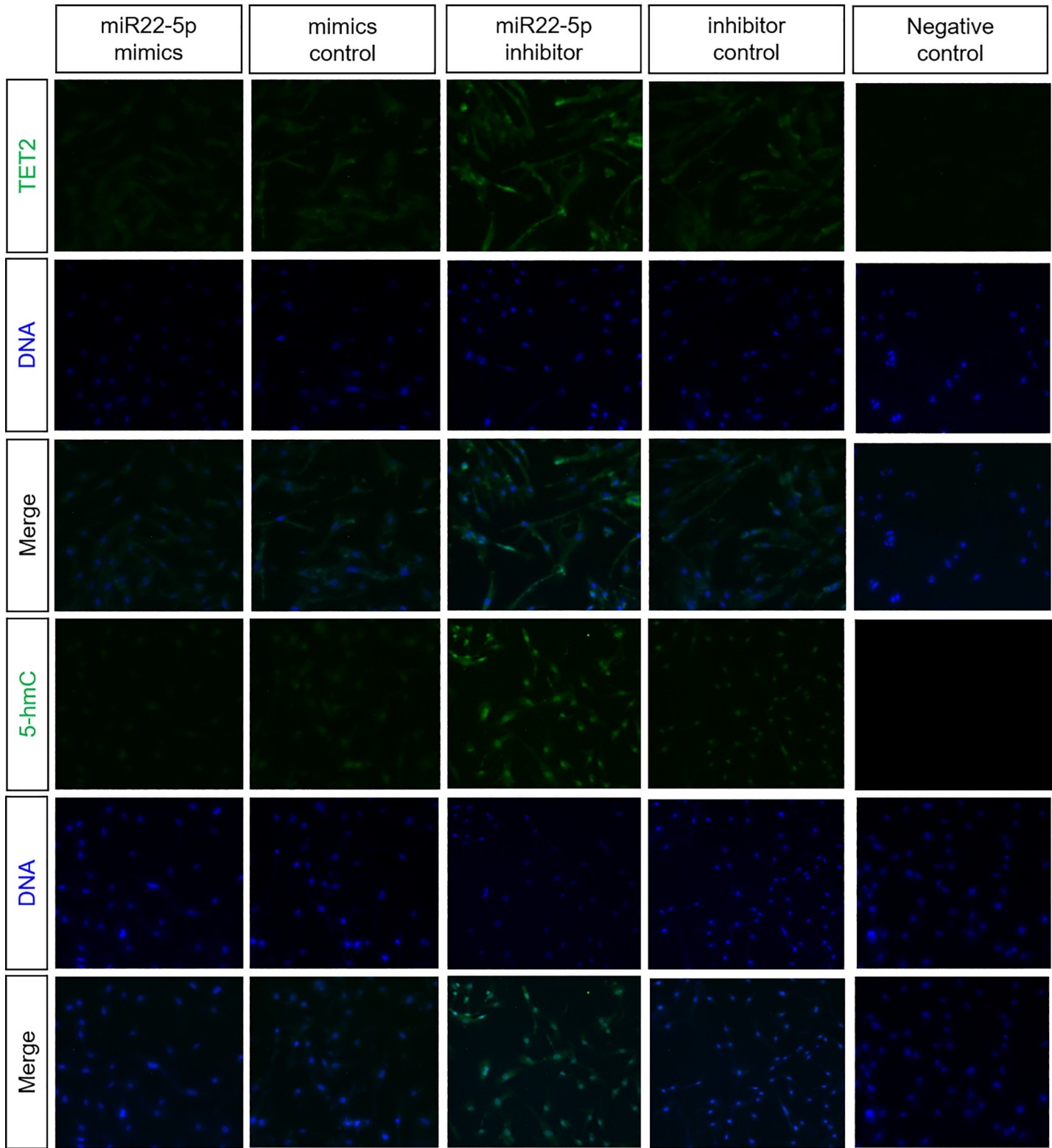

**Fig 3. Immunofluorescence expression of TET2 and 5-hmC in ESCs following treatment with miR22-5p mimics and inhibitor.**

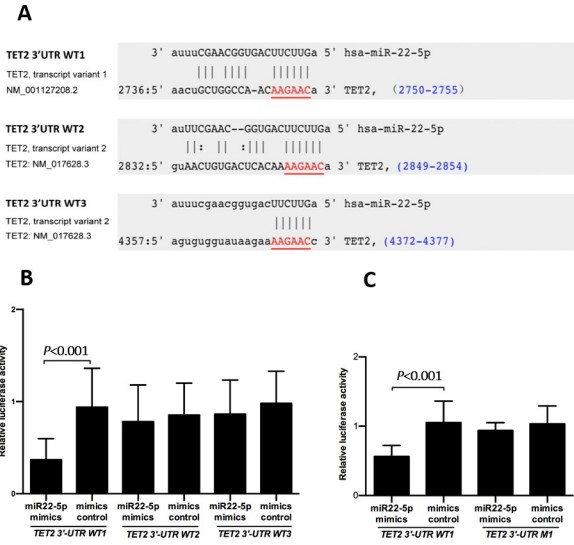

**Fig 4. MiR22-5p directly regulates the *TET2* 3′-UTR.** (A) The *TET2* 3′-UTR contains three possible miR22-5p-binding sites. The possible TET2-binding sites are indicated in red. (B) Relative luciferase activity for the three possible WT binding sites (n = 3). (C) Luciferase activity for TET2 3′-UTR WT1 and M1 (n = 3). Data were analyzed by Student's *t*-test.

tumorigenesis, and represents a tumorigenic pathway in addition to the more familiar TET family mutations and deletions [24, 25]. These studies indicated that the regulatory relationships between TET2 and miR-22 or miR22-5p are worth investigating in other tissues. Based on a comprehensive analysis of minimal/mild endometriosis patients, we found that TET2 is upregulated in the endometrium during implantation window, and is directly affected by the miR22-5p level.

A subset of DNA hypomethylated canyons is maintained by the cooperative action of TET proteins, in particular, TET1 and TET2 [26]. Aberrant expression of epigenetic alterations in endometriosis include genomic DNA methylation of the gene encoding progesterone receptor-β [27], *HOXA10* [28], *ESR2* [13], which are candidate genes responsible for the development of progesterone resistance and implantation failure. The *ESR2* mRNA level was significantly increased in endometriotic ESCs when compared to normal ESCs. A hypomethylated (−197/+359) promoter region of *ESR2* in endometriotic cells was previously considered the primary mechanism responsible for the differential ESR2 expression in endometriotic and normal endometrium [13]. We investigated whether increased TET2 expression due to decreased miR22-5p expression affected ESR2 expression, and we found that ESCs transfected with miR22-5p mimics demonstrated significantly attenuated ESR2 expression. Bioinformatics and genetic analyses revealed no potential miR22-5p target sites in the 3′-UTR of the *ESR2* mRNA sequence. Indeed, it is speculated that miR22-5p may regulate *ESR2* promoter methylation by directly targeting *TET2*. Bisulfite sequencing of this promoter region (−197/+359) showed that there was no significant difference in the methylation status after transfection with miR22-5p inhibitor. Methylation tended to be lower in miR22-5p inhibitor-transfected cells than in control cells, but the promoter was extremely hypomethylated in both cases, which was inconsistent with a previous study [13]. In Xue's study, eutopic endometrium was not compared between subjects with endometriosis and disease-free subjects, and the menstrual cycles of the study subjects were unknown [13], whereas we utilized mid-luteal endometrium of infertile women with minimal/mild endometriosis in the embryo implantation phase.

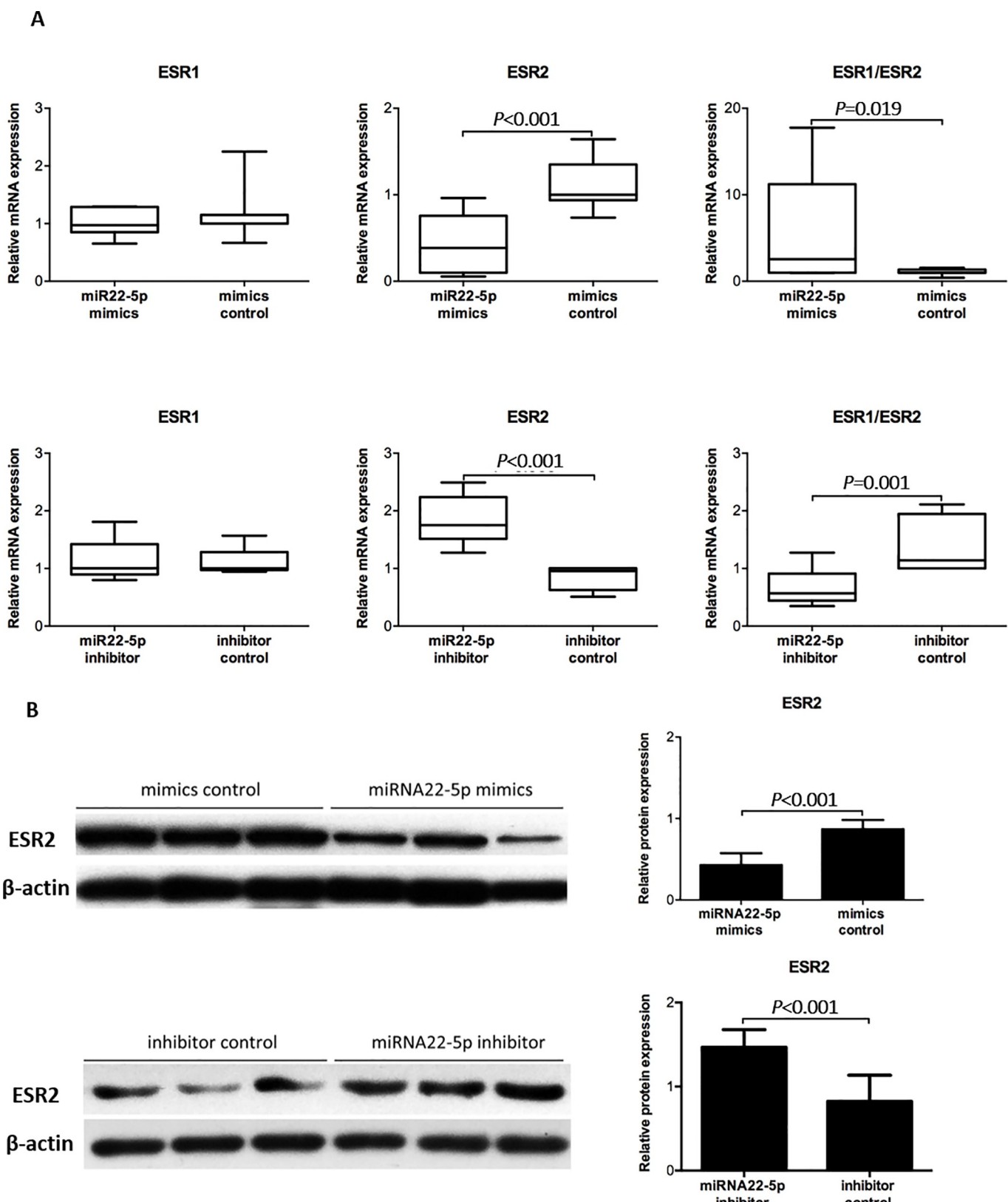

**Fig 5. MiR22-5p affects ESR1 and ESR2 expression.** (A) RT-qPCR analysis of mRNA expression of *ESR1* and *ESR2* and the *ESR1/ESR2* mRNA ratio following treatment of ESCs with miR22-5p mimics and inhibitor (n = 3). (B) Western blot analysis of ESR2 protein levels following the treatment of ESCs with miR22-5p mimics and inhibitor (n = 3). Data were analyzed by Student's *t*-test.

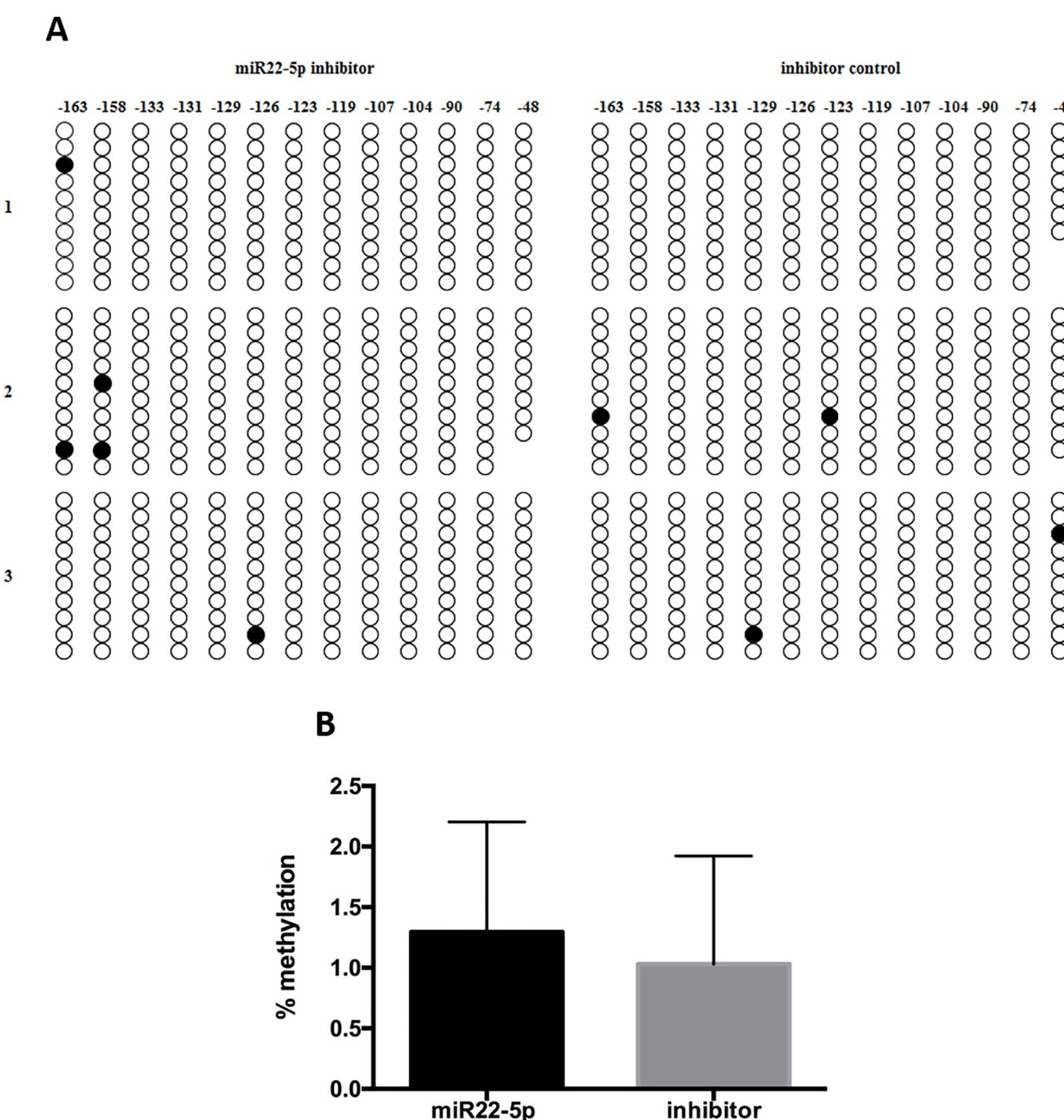

**Fig 6. DNA methylation status of the *ESR2* promoter region (−197/+359) in ESCs following treatment with miR22-5p mimics and inhibitor.** (A) Methylation status of 13 CpG sites in the *ESR2* promoter region after treatment with miR22-5p inhibitor as indicated by bisulfite sequencing (n = 3). The numbers indicate the positions of cytosine residues of CpGs relative to the transcription start site (+1). (B) Percent methylation of the *ESR2* promoter region in ESCs after treatment with miR22-5p inhibitor. Data were analyzed by Student's *t*-test.

DNA methylation profiles of human endometrium differ in different phases of the cycle [29, 30], which could explain the opposing findings in our and other studies. Overall, our findings suggested that miR22-5p directly dysregulates TET2, which modified the 5-hmC level and

altered ESR2, but not ESR1 expression in ESCs. This also contributed to an increase in the *ESR1/ESR2* mRNA ratio. ESR2 is suggested to modulate ESR1 activity; thus, a change in the relative expression levels of ESR2 to ESR1 suggests a differential regulatory response in estrogen signaling [13]. PR expression in the secretory phase of the menstrual cycle is regulated through ESR1 [31]. Therefore, altered miR22-5p expression in the endometrium may lead to dysregulation of the progesterone response and consequently, implantation-related infertility in women with endometriosis.

To our knowledge, this is the first study to demonstrate downregulation of miR22-5p in eutopic endometrium of women with endometriosis during the secretory phase. Further, we showed that TET2 upregulation during implantation window in human endometrium was directly regulated by miR22-5p. MiR22-5p, which has not been thoroughly studied in endometriosis to date, directly dysregulated the expression of TET2, which is a key marker for DNA hydroxymethylation.

There are some limitations of our study. The first is that we found that ESR2 was also upregulated by miR22-5p, but it was not through the regulation of promoter region methylation of ESR2; hence, the regulatory mechanisms are needed to be further studied. In addition, some studies have revealed the functions of miR22-5p in other diseases [32, 33]. However, this study only focused on that miR22-5p dysregulates directly the expression of TET2 in the eutopic endometrium of endometriosis without the exploration of specific functions of miR22-5p. The role of miR22-5p in modulating the function of the eutopic ESCs and endometrial receptivity in endometriosis also requires exploration in the following study.

## Supporting information

**S1 Raw images.**
(PDF)

## Acknowledgments

We thank Dr. Shanti Gurung from The Ritchie Centre, Hudson Institute of Medical Research for language modification.

## Author Contributions

**Conceptualization:** Li Xiao.

**Data curation:** Li Xiao, Wei Huang, Jing Tan.

**Formal analysis:** Li Xiao, Min Zhou, Jing Tan, Tingting Liu.

**Funding acquisition:** Wei Huang.

**Investigation:** Li Xiao, Tianjiao Pei, Wei Huang, Min Zhou, Jing Fu, Jing Tan, Tingting Liu, Yong Song, Shiyuan Yang.

**Methodology:** Li Xiao, Min Zhou, Jing Fu, Yong Song, Shiyuan Yang.

**Supervision:** Wei Huang.

**Validation:** Li Xiao, Tianjiao Pei.

**Writing – original draft:** Li Xiao.

**Writing – review & editing:** Li Xiao, Wei Huang.

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
