## [Decision Letter · Decision Letter 0]

21 Jan 2020

PONE-D-19-32583

MicroRNA22-5p targets TET2 and regulates Estrogen receptor 2 expression in infertile women with minimal/mild endometriosis during implantation window

PLOS ONE

Dear Prof. Huang,

Thank you for submitting your manuscript to PLOS ONE. After careful consideration, we feel that it has merit but does not fully meet PLOS ONE’s publication criteria as it currently stands. Therefore, we invite you to submit a revised version of the manuscript that addresses the points raised during the review process.

We would appreciate receiving your revised manuscript by Mar 06 2020 11:59PM. To enhance the reproducibility of your results, we recommend that if applicable you deposit your laboratory protocols in protocols.io, where a protocol can be assigned its own identifier (DOI) such that it can be cited independently in the future. For instructions see: http://journals.plos.org/plosone/s/submission-guidelines#loc-laboratory-protocols

We look forward to receiving your revised manuscript.

Kind regards,

Jae-Wook Jeong, Ph.D.

Academic Editor

PLOS ONE

2. We noticed you have some minor occurrence(s) of overlapping text with the following previous publication(s), which needs to be addressed:

https://doi.org/10.1093/humrep/dew223

https://doi.org/10.1095/biolreprod.107.061804

https://doi.org/10.1186/1477-7827-12-42

https://doi.org/10.1210/jc.2012-3010

https://doi.org/10.1210/en.2018-00374

https://doi.org/10.1186/s12958-017-0234-9

https://doi.org/10.1016/j.stem.2013.06.003

https://doi.org/10.1007/s00441-015-2339-9

In your revision ensure you cite all your sources (including your own works), and quote or rephrase any duplicated text outside the Methods section. Further consideration is dependent on these concerns being addressed.

3. We note that you have reported significance probabilities of 0 in places. Since p=0 is not strictly possible, please correct this to a more appropriate limit, eg 'p<0.0001'.

4. Please include in your Methods section the date ranges over which you recruited participants to this study.

Reviewers' comments:

Reviewer's Responses to Questions

**Comments to the Author**

1. Is the manuscript technically sound, and do the data support the conclusions?

Reviewer #1: Yes

Reviewer #2: Partly

2. Has the statistical analysis been performed appropriately and rigorously? 

Reviewer #1: Yes

Reviewer #2: I Don't Know

3. Have the authors made all data underlying the findings in their manuscript fully available?

Reviewer #1: Yes

Reviewer #2: Yes

4. Is the manuscript presented in an intelligible fashion and written in standard English?

Reviewer #1: Yes

Reviewer #2: Yes

5. Review Comments to the Author

Reviewer #1: In this manuscript, the authors showed that MicroRNA22-5p targets TET2 and regulates Estrogen receptor 2 expression in infertile women with minimal/mild endometriosis during implantation window. The authors suggest the role of miR22-5p in endometriosis-associated infertility and the significant findings from this study were that a decrease of miR22-5p and an increase of TET2 expression were observed in minimal/mild endometriosis during implantation window. TET2 was a key directly target of miR22-5p and MiR22-5p regulated the expression of ESR2. The topic of the paper is interesting although some minor concerns have been raised after carefully reading the work.

1. Is “disease-free women” group infertile or fertile?

2. What do you mean “another 13 mild/minimal endometriosis and 11 normal controls” ? Are these group from 24 of normal and 26 of endometriosis in “Study Population “ section? The authors need to provide the detail human sample information.

3. The authors described that all participants were determined to be in the proliferative or secretory phase. Is there any different expression between proliferative and secretory phase in Figure 1A?

4. Again, the author only used secretory phase group for Figure 1B. The author needs to clarify the expression between proliferative and secretory phase.

5. The authors need to provide high magnification picture for better understanding in Figure 1D.

6. The statistical analysis should be t-test not ANOVA in Figure 1.

7. What is the rationale to use primary ESCs cell? TET2 are highly expressed in stromal and epithelial cells in endometriosis group.

8. What primary cells were used for Figure 5? The authors need to be clarified.

9. The authors described that ESR2 represses ESR1 gene expression by directly binding to ESR1 promoter region, which can result in a state of progesterone resistance but the expression of ESR1 were not changed in Figure 5. What is the author’s conclusion?

10. How was the expression of ESR1 by Western blot in Figure 5?

11. As TET2 are expressed in stromal and epithelial cells, it will be helpful if the authors add the result in epithelial cells for depth of the authors’ conclusion.

Reviewer #2: This study was to investigate the expression and role of a microRNA, miR22-5p, in endometriosis. The authors previously found that miR22-5p was downregulated in the endometrium of patients with mild endometriosis. The current study confirmed this finding and further explored its role using primary endometrial stromal cells and 293T cells. The authors showed that TET2 was a direct target gene of miR22-5p in endometrial cells, and ESR2 expression was also affected by manipulation of miR22-5p levels through an unidentified mechanism. The experiments were overall well described. However, there are several major concerns on the rigor, design, and language that need to be addressed.

Major concerns:

1. A major weakness of this manuscript is that there are no functional studies designed to evaluate the role of miR22-5p. It is necessary to include cell culture experiments to address the role of miR22-5p in endometrial stromal cell properties.

2. The authors stated that normal endometrial tissues were obtained from 24 disease-free women. However, it was unclear under what conditions these women underwent surgeries.

3. There are no negative controls (i.e. the use of isotype matched IgGs controls) for immunofluorescence and immunohistochemistry.

4. The authors indicated that 6 experiments were performed for Luciferase Reporter Gene Assay. However, results in Fig. 4 showed n = 3. Are only 3 experiments out of 6 showed or n refers to 3 technical replicates?

5. The manuscript contains many grammatical errors and vague statements. A thorough language editing appears necessary.

Minor concerns:

1. ANOVA (under Fig. 1 legend) was not mentioned in the statistical analysis section. In addition, why ANOVA was used in these experiments as only two means were compared?

2. Descriptions of sample numbers (n) are missing in Fig. 5 and Fig. 6.

3. Fig. 5: ESR1 protein levels should also be examined and shown.

4. The TET2 result in mimics control and miR22-5p mimics group in Fig. 2F seems inconsistent with the quantification result listed in panel E.

5. There is no justification for using 293T cells in this study. Figure legend

6. PLOS authors have the option to publish the peer review history of their article (what does this mean?). If published, this will include your full peer review and any attached files.

Reviewer #1: No

Reviewer #2: No

---

## [Author Response · Author response to Decision Letter 0]

1 Mar 2020

（1）Responses to editors

Reply: Thank you for your reminding. The revised manuscript has been checked and meets PLOS ONE’s style requirement. 

2. We noticed you have some minor occurrence(s) of overlapping text with the following previous publication(s), which needs to be addressed:

https://doi.org/10.1093/humrep/dew223

https://doi.org/10.1095/biolreprod.107.061804

https://doi.org/10.1186/1477-7827-12-42

https://doi.org/10.1210/jc.2012-3010

https://doi.org/10.1210/en.2018-00374

https://doi.org/10.1186/s12958-017-0234-9

https://doi.org/10.1016/j.stem.2013.06.003

https://doi.org/10.1007/s00441-015-2339-9

In your revision ensure you cite all your sources (including your own works), and quote or rephrase any duplicated text outside the Methods section. Further consideration is dependent on these concerns being addressed.

Reply: 

For https://doi.org/10.1093/humrep/dew223, it is “miR-196a overexpression activates the MEK/ERK signal and represses the progesterone receptor and decidualization in eutopic endometrium from women with endometriosis”. This article was one of our previous publications about miRNAs research on endometriosis. We found 66 differentially expressed miRNAs, and miR22-5p was one of those from this study. It was Ref.15, which has been quoted except in Methods sections.

For https://doi.org/10.1095/biolreprod.107.061804, it is “Promoter Methylation Regulates Estrogen Receptor 2 in Human Endometrium and Endometriosis”, which was the first one to determine the hypomethylated promoter of ESR2 gene, which was the basis of our study. It was Ref. 13, which has been quoted except in Methods sections. 

For https://doi.org/10.1186/1477-7827-12-42, it is “Increased expression of the pluripotency markers sex-determining region Y-box 2 and Nanog homeobox in ovarian endometriosis”, which is one of our previous publications. The experimental procedures of “RNA extraction and Quantitative Real-time PCR”, “Western Blot Analysis”, and “Immunohistochemistry” used in this study were modified partly from this article.

For https://doi.org/10.1210/jc.2012-3010, it is “MicroRNA23a and MicroRNA23b Deregulation Derepresses SF-1 and Upregulates Estrogen Signaling in Ovarian Endometriosis”, which was the first article from our group working on miRNAs, published in 2013. It was Ref. 17, which has been quoted except in Methods sections.

For https://doi.org/10.1210/en.2018-00374, it is “miR-194-3p Represses the Progesterone Receptor and Decidualization in Eutopic Endometrium from Women with Endometriosis”, which also focused on the miRNAs on endometriosis. It was also a following research of that original research. It was Ref. 16, which has been quoted except in Methods sections.

For https://doi.org/10.1186/s12958-017-0234-9, it is “Expression of SOX2, NANOG and OCT4 in a mouse model of lipopolysaccharide-induced acute uterine injury and intrauterine adhesions”, which is one of our previous publications. The experimental procedures of “RNA extraction and Quantitative Real-time PCR”, “Western Blot Analysis”, and “Immunohistochemistry” used in this study were also modified partly from this article.

For https://doi.org/10.1016/j.stem.2013.06.003, it is “The oncogenic microRNA miR-22 Target the TET2 tumor suppressor to promote hematopoietic stem cell self-renewal and transformation”. It was the first research to identify that miR-22 contributes to inactivation of TET2 and other TET family members in tumorigenesis. It was Ref. 24, which has been quoted except in Methods sections.

For https://doi.org/10.1007/s00441-015-2339-9, it is “Oxidative stress and oocyte quality: ethiopathogenic mechanisms of minimal/mild endometriosis-related infertility”. In the introduction section of our manuscript, the introduction of the mechanisms of minimal/mild endometriosis was quoted from this paper, the revised manuscript has been carefully modified and cited as Ref. 4.

3. We note that you have reported significance probabilities of 0 in places. Since p=0 is not strictly possible, please correct this to a more appropriate limit, eg 'p<0.0001'.

Reply: In the revised manuscript, all those “p=0” were corrected to “p=0.000“. Those changes could be found in the revised Figures (Fig 1, Fig 2, Fig 4 and Fig 5) and Page 14, line 21-26; Page 16, line 9-18, which have been highlighted in manuscript. 

4. Please include in your Methods section the date ranges over which you recruited participants to this study.

Reply: The date has been added to the study population section on Page 6, line 1-3 in the revised manuscript with highlight.

Reply: Our blot/gel image data had been provided in Supporting Information in the resubmission.

(2) Responses to reviewers

Reviewer #1: In this manuscript, the authors showed that MicroRNA22-5p targets TET2 and regulates Estrogen receptor 2 expression in infertile women with minimal/mild endometriosis during implantation window. The authors suggest the role of miR22-5p in endometriosis-associated infertility and the significant findings from this study were that a decrease of miR22-5p and an increase of TET2 expression were observed in minimal/mild endometriosis during implantation window. TET2 was a key directly target of miR22-5p and MiR22-5p regulated the expression of ESR2. The topic of the paper is interesting although some minor concerns have been raised after carefully reading the work.

1. Is “disease-free women” group infertile or fertile?

Reply: The patients from disease-free women group were infertile. All patients recruited in this study were infertile. We have added the detailed description of patients on page 6, line 1-4.

2. What do you mean “another 13 mild/minimal endometriosis and 11 normal controls”? Are these group from 24 of normal and 26 of endometriosis in “Study Population “section? The authors need to provide the detail human sample information.

Reply: 50 infertile patients were recruited in this study that 24 patients with normal endometrium as control and 26 patients who suffered minimal/mild endometriosis, which all samples were from secretory phase. During the 24 control patients, 11 samples were used for PCR analysis, 5 samples were used for Western Blot, 3 samples were for IHC analysis, and 6 samples were for cell experiment. During the 26 endometriosis patients, 13 samples were used for PCR analysis, 6 samples were used for Western Blot, 3 samples were for IHC analysis, and 6 samples were for cell experiment. The details of human sample information have been attached to the revised submission.

3. The authors described that all participants were determined to be in the proliferative or secretory phase. Is there any different expression between proliferative and secretory phase in Figure 1A?

Reply: As this study focused on the changes of implantation window, the expression or analysis of the data was from the endometrial samples of secretory phase. All samples reported and used in the present study were from secretory phase which was determined by the timing of the patients’ last menstrual period and histological analysis of endometrium. As interested, we did have some IHC staining of proliferative phase, which is as follow. For the purpose of the study, it may be no need to include patients from proliferative phase.

4. Again, the author only used secretory phase group for Figure 1B. The author needs to clarify the expression between proliferative and secretory phase.

Reply: As our previous microRNA microarray data in mid-luteal endometrium showed that the expression of miR22-5p decreased, the aim of this study was to investigate the effects/biological functions of miR22-5p in minimal/mild endometriosis during secretory phase. Therefore, all data and analysis in this study were in the secretory phase, the comparison between proliferative and secretory phase has not been investigated yet in the present study, which we will further focus on it in future studies.

5. The authors need to provide high magnification picture for better understanding in Figure 1D.

Reply: A higher magnification picture of Figure 1D has been added in Figure 1.

6. The statistical analysis should be t-test not ANOVA in Figure 1.

Reply: Thanks for your attention. The data has been re-analyzed by t-test, the change was highlighted in the figure legend of Figure 1.

7. What is the rationale to use primary ESCs cell? TET2 are highly expressed in stromal and epithelial cells in endometriosis group.

Reply: Primary cells are directly derived from body tissues, and their biological characteristics have not changed significantly, so they can reflect the state of the disease to a certain extent. Our group has established a stable method for the isolation and culture of primary endometrial stromal cells for several years. As primary endometrial epithelial cell could not be passage and sub-cultured, we used primary endometrial stromal cell for Primary Cell Culture and Transfection. 

8. What primary cells were used for Figure 5? The authors need to be clarified.

Reply: Primary endometrial stromal cells (ESCs) were used for Figure 5. The change has been highlighted in Figure legend of Fig 5 on Page 16, Line 22.

9. The authors described that ESR2 represses ESR1 gene expression by directly binding to ESR1 promoter region, which can result in a state of progesterone resistance but the expression of ESR1 were not changed in Figure 5. What is the author’s conclusion?

Reply: In previous research (Bulun SE, et al. Semin Reprod Med, 2012), they found ESR2 represses ESR1 gene expression by directly binding to ESR1 promoter region in endometriotic tissue, which can result in a state of progesterone resistance. But in our study, there was no significant change in the expression of ESR1 after the transfection of miR22-5p mimics and inhibitor in eutopic endometrium. ESR2 may only repress ESR1 gene expression in some specific situation and location, or the expression of ESR1 has been affected by other unknown ways after transfection of miR22-5p mimics and inhibitor, which may be worthy to further investigate.

10. How was the expression of ESR1 by Western blot in Figure 5?

Reply: There was no significant difference of ESR1 mRNA level in our study, analyzed by qRT-PCR. As knowing about how miRNAs work , and we have a standard protocol of qRT-PCR in miRNAs research since 2013 (Ref.17, Shen, et al. JCEM, 2013), the protein expression was not that crucial and necessary when no difference in mRNA level after miRNAs interfered, so ESR1 expression by WB was not performed when we did the experiments. As reviewer’s interest, we are also trying to do a WB for ESR1, but we do have very rare infertile patients visiting hospital, due to the Covid-19, which is spreading in China recently.

11. As TET2 are expressed in stromal and epithelial cells, it will be helpful if the authors add the result in epithelial cells for depth of the authors’ conclusion.

Reply: We do agree with this opinion. We were also intending to investigate the expression of TET2 in endometrial epithelial cells, but failed to do the primary endometrial epithelial cell passage and sub-culture. The primary endometrial epithelial cell passage and sub-culture are still an unsolved problem for endometrial related researches. If solved, it will greatly contribute to the investigations of endometrial diseases.

Reviewer #2: This study was to investigate the expression and role of a microRNA, miR22-5p, in endometriosis. The authors previously found that miR22-5p was downregulated in the endometrium of patients with mild endometriosis. The current study confirmed this finding and further explored its role using primary endometrial stromal cells and 293T cells. The authors showed that TET2 was a direct target gene of miR22-5p in endometrial cells, and ESR2 expression was also affected by manipulation of miR22-5p levels through an unidentified mechanism. The experiments were overall well described. However, there are several major concerns on the rigor, design, and language that need to be addressed.

Major concerns:

1. A major weakness of this manuscript is that there are no functional studies designed to evaluate the role of miR22-5p. It is necessary to include cell culture experiments to address the role of miR22-5p in endometrial stromal cell properties.

Reply: Thanks to this comment. We also tried to figure out the role of miR22-5p in endometriosis related infertility. We performed in vitro decidualization experiment to investigate the effect of miR22-5p on progesterone resistance which our group focused on all along and reported previously (Min Zhou, at el, 2017, and Tianjiao Pei, et al. 2018, Ref 14 and 15). Unfortunately, miR22-5p downregulation did not promote or hinder decidualization in ESCs, which is showing as follow. But we didn’t put this result in our manuscript.

We will keep on further investigating the role of miR22-5p, as well as TET2, in endometriosis related infertility by other functional studies.

2. The authors stated that normal endometrial tissues were obtained from 24 disease-free women. However, it was unclear under what conditions these women underwent surgeries.

Reply: We were very sorry to confuse the reviewers. All patients recruited in our study were infertile. Laparoscopy and hysteroscopy were performed, those 24 women suffered from peritubal adhesion, mesosalpinx cyst or pelvic adhesion. Normal endometrial tissues were obtained from those 24 infertile women without any endometrial pathology confirmed by hysteroscopy. 

3. There are no negative controls (i.e. the use of isotype matched IgGs controls) for immunofluorescence and immunohistochemistry.

Reply: Sorry for our negligence. Images of negative controls have been added to Fig.1 for Immunohistochemistry and Fig. 3 for Immunofluorescence. The isotype controls were Rabbit lgG and Mouse IgG. Changes have been highlighted on Page 11, Line 14-15, also, presenting in Fig.1 and Fig. 3

4. The authors indicated that 6 experiments were performed for Luciferase Reporter Gene Assay. However, results in Fig. 4 showed n = 3. Are only 3 experiments out of 6 showed or n refers to 3 technical replicates?

Reply: Thanks for figuring out this. The Luciferase Reporter Gene Assay experiments were repeated three times with six technical replicates for wild-type TET2 3’-UTRs and mutation of TET2 3’-UTR WT1 each. The change has been highlighted on Page 10, Line 3.

5. The manuscript contains many grammatical errors and vague statements. A thorough language editing appears necessary.

Reply: We have carefully reviewed and re-edited the whole manuscript. Dr. Shanti Gurung from The Ritchie Centre, Hudson Institute of Medical Research, Melbourne, Australia, greatly contributes to edit the revised manuscript. We are grateful for her efforts and help in language modification which is presenting in Acknowledge session. The changes have been highlighted in revised manuscript.

Minor concerns:

1. ANOVA (under Fig. 1 legend) was not mentioned in the statistical analysis section. In addition, why ANOVA was used in these experiments as only two means were compared?

Reply: Thanks for kind reminding this inappropriate statistical analysis. The data has been re-analyzed by Student’s t test for the comparison for two groups, the change was highlighted in the figure legend of Fig. 1(Page 14, Line 10-11).

2. Descriptions of sample numbers (n) are missing in Fig. 5 and Fig. 6.

Reply: Sorry for the omission. The sample number for Primary Cell Culture and Transfection was three in Fig. 2, Fig.5 and Fig.6, which have been highlighted on Page 6, Line 13, Page 16, Line 22 and Page 17, Line 13.

3. Fig. 5: ESR1 protein levels should also be examined and shown.

Reply: Thanks very much for this concern. In our present study, there was no significant difference of ESR1 mRNA level, analyzed by qRT-PCR. As knowing about how miRNAs work , and we have a standard protocol of qRT-PCR in miRNAs research since 2013 (Ref.17, Shen, et al. JCEM, 2013), the protein expression was not that crucial and necessary when no difference in mRNA level after miRNAs inferfered, so ESR1 expression by WB was not performed when we did the experiments previously. As reviewer’s interest, we were also trying to do a WB for ESR1, but we do have very rare infertile patients visiting hospital recently, due to the Covid-19, which is spreading in China. If a ESR1 expression of WB have been required 

4. The TET2 result in mimics control and miR22-5p mimics group in Fig. 2F seems inconsistent with the quantification result listed in panel E.

Reply: We are very sorry to confuse the reviewer. The word “TET2” under Fig. 2F was for Fig. 2E, which was presenting the analysis of TET2 expression after transfection of miRNA22-5p by WB. Fig. 2F was showing the expression of 5-hmC expression by Dot blot. TET2 is responsible for conversion of 5-mC in to 5-hmC, therefore, 5-hmC would be unregulated by the down-expressed TET2 when transfecting with miR22-5p mimics, the description was showing on Page 15, Line 1-3.

5. There is no justification for using 293T cells in this study. Figure legend

Reply: 293T cell is a derivative of 293 cells that stably express the large T-antigen of SV40. The presence of T-antigen in 293T cell could help in episomal maintenance of SV40 origin containing vectors. High-density transfection of 293T cells allows doubling of transient titers and remove need for a priori DNA complex formation with PEI. In case of transfection, 293T cell is now preferred since they are normally more transfectable. On the other hand, the primary endometrial stromal cells are not stable enough for Luciferase Reporter Gene Assay to avoid the influence of other factors. Therefore, 293T cells is the most common used for Luciferase Reporter Gene Assay.

6. PLOS authors have the option to publish the peer review history of their article (what does this mean?). If published, this will include your full peer review and any attached files.

Reply: we agree to publish the peer review history of our article.

---

## [Decision Letter · Decision Letter 1]

16 Mar 2020

PONE-D-19-32583R1

MicroRNA22-5p targets TET2 and regulates Estrogen receptor 2 expression in infertile women with minimal/mild endometriosis during implantation window

PLOS ONE

Dear Prof. Huang,

Thank you for submitting your manuscript to PLOS ONE. After careful consideration, we feel that it has merit but does not fully meet PLOS ONE’s publication criteria as it currently stands. Therefore, we invite you to submit a revised version of the manuscript that addresses the points raised during the review process.

We would appreciate receiving your revised manuscript by Apr 30 2020 11:59PM. To enhance the reproducibility of your results, we recommend that if applicable you deposit your laboratory protocols in protocols.io, where a protocol can be assigned its own identifier (DOI) such that it can be cited independently in the future. For instructions see: http://journals.plos.org/plosone/s/submission-guidelines#loc-laboratory-protocols

We look forward to receiving your revised manuscript.

Kind regards,

Jae-Wook Jeong, Ph.D.

Academic Editor

PLOS ONE

Reviewers' comments:

Reviewer's Responses to Questions

**Comments to the Author**

1. If the authors have adequately addressed your comments raised in a previous round of review and you feel that this manuscript is now acceptable for publication, you may indicate that here to bypass the “Comments to the Author” section, enter your conflict of interest statement in the “Confidential to Editor” section, and submit your "Accept" recommendation.

Reviewer #1: All comments have been addressed

Reviewer #2: (No Response)

2. Is the manuscript technically sound, and do the data support the conclusions?

Reviewer #1: Yes

Reviewer #2: Yes

3. Has the statistical analysis been performed appropriately and rigorously? 

Reviewer #1: Yes

Reviewer #2: Yes

4. Have the authors made all data underlying the findings in their manuscript fully available?

Reviewer #1: Yes

Reviewer #2: Yes

5. Is the manuscript presented in an intelligible fashion and written in standard English?

Reviewer #1: Yes

Reviewer #2: No

6. Review Comments to the Author

Reviewer #1: The authors have addressed comments from the initial review. This manuscript is appropriate for publication.

Reviewer #2: The authors are commended for their efforts to address the reviewer’s comments. However, there are some concerns remain.

Major concern # 1: Since the authors know nothing about the function of miR22-5p, it is suggested that this limitation be discussed in the “discussion section”.

Major concern # 6: The language has been improved, but there are still many errors and awkward sentences that prevent a clear understanding of the manuscript. Language editing is strongly recommended. A few examples are listed below:

Page 2, Line 15-16: MiR22-5p regulated the expression of ESR2, but do not directly affect the methylation of ESR2 promoter region. In this sentence, “do not” should be “did not”.

Page 2, Lines 17-18: This study provides a novel approach for the imbalance of miR22-5p expression in mid-luteal endometrium of minimal/mild endometriosis, may involve in the mechanism of its associated infertility. This sentence is grammatically incorrect.

Page 21, Lines 20-21: Our findings may provide a new novel approach to target in the mechanisms of minimal/mild endometriosis associated infertility. In this sentence, new and novel are repetitive. The sentence is also grammatically inappropriate.

7. PLOS authors have the option to publish the peer review history of their article (what does this mean?). If published, this will include your full peer review and any attached files.

Reviewer #1: No

Reviewer #2: No

---

## [Author Response · Author response to Decision Letter 1]

7 May 2020

Reviewer #2: The authors are commended for their efforts to address the reviewer’s comments. However, there are some concerns remain.

Major concern # 1: Since the authors know nothing about the function of miR22-5p, it is suggested that this limitation be discussed in the “discussion section”.

Reply: The limitation of this study has been discussed in the “discussion section”, which is showing on Page 21, Line 3-11.

Major concern # 6: The language has been improved, but there are still many errors and awkward sentences that prevent a clear understanding of the manuscript. Language editing is strongly recommended. A few examples are listed below:

Page 2, Line 15-16: MiR22-5p regulated the expression of ESR2, but do not directly affect the methylation of ESR2 promoter region. In this sentence, “do not” should be “did not”.

Reply: Thanks for figuring out this inappropriate grammar. “do not” has been replaced with “did not”, which is showing on Page2, Lines 15-16.

Page 2, Lines 17-18: This study provides a novel approach for the imbalance of miR22-5p expression in mid-luteal endometrium of minimal/mild endometriosis, may involve in the mechanism of its associated infertility. This sentence is grammatically incorrect.

Reply: Thanks for this concern. This sentence has been reorganized in the revised manuscript, and it is showing on Page 2, Lines 17-18. 

Page 21, Lines 20-21: Our findings may provide a new novel approach to target in the mechanisms of minimal/mild endometriosis associated infertility. In this sentence, new and novel are repetitive. The sentence is also grammatically inappropriate.

 Reply: Thanks for this reminding, “new” has been removed in the revision, which could be found on Page 21, Lines 20-21.

Thanks for the comments on the language. The revised manuscript has been reedited by Editage services (http://app.editage.com/). We are hoping that English writing in this revision has been improved and meet the required standards for publication.

---

## [Decision Letter · Decision Letter 2]

19 May 2020

MicroRNA22-5p targets ten-eleven translocation and regulates estrogen receptor 2 expression in infertile women with minimal/mild endometriosis during implantation window

PONE-D-19-32583R2

Dear Dr. Huang,

We are pleased to inform you that your manuscript has been judged scientifically suitable for publication and will be formally accepted for publication once it complies with all outstanding technical requirements.

With kind regards,

Jae-Wook Jeong, Ph.D.

Academic Editor

PLOS ONE

Additional Editor Comments (optional):

Reviewers' comments:

Reviewer's Responses to Questions

**Comments to the Author**

1. If the authors have adequately addressed your comments raised in a previous round of review and you feel that this manuscript is now acceptable for publication, you may indicate that here to bypass the “Comments to the Author” section, enter your conflict of interest statement in the “Confidential to Editor” section, and submit your "Accept" recommendation.

Reviewer #2: All comments have been addressed

2. Is the manuscript technically sound, and do the data support the conclusions?

Reviewer #2: Yes

3. Has the statistical analysis been performed appropriately and rigorously? 

Reviewer #2: I Don't Know

4. Have the authors made all data underlying the findings in their manuscript fully available?

Reviewer #2: Yes

5. Is the manuscript presented in an intelligible fashion and written in standard English?

Reviewer #2: Yes

6. Review Comments to the Author

Reviewer #2: Page 21, line 3 &4: Authors stated that “…we found that ESR2 was also upregulated by miR22-5p”. This statement conflicts with the results that “ESR2 mRNA expression was significantly downregulated after transfection with miR22-5p mimics (page 16, lines 6&7).”

7. PLOS authors have the option to publish the peer review history of their article (what does this mean?). If published, this will include your full peer review and any attached files.

Reviewer #2: No

---

## [Editor Report · Acceptance letter]

15 Jun 2020

PONE-D-19-32583R2 

MicroRNA22-5p targets ten-eleven translocation and regulates estrogen receptor 2 expression in infertile women with minimal/mild endometriosis during implantation window 

Dear Dr. Huang:

I'm pleased to inform you that your manuscript has been deemed suitable for publication in PLOS ONE. Congratulations! Your manuscript is now with our production department. 

Kind regards, 

on behalf of

Prof. Jae-Wook Jeong 

Academic Editor

PLOS ONE